# Extracellular Matrix as a Target in Melanoma Therapy: From Hypothesis to Clinical Trials

**DOI:** 10.3390/cells13221917

**Published:** 2024-11-19

**Authors:** Yuriy P. Mayasin, Maria N. Osinnikova, Chulpan B. Kharisova, Kristina V. Kitaeva, Ivan Y. Filin, Anna V. Gorodilova, Grigorii I. Kutovoi, Valeriya V. Solovyeva, Anatolii I. Golubev, Albert A. Rizvanov

**Affiliations:** 1Institute of Fundamental Medicine and Biology, Kazan Federal University, 420008 Kazan, Russia; mayasin_yuriy@mail.ru (Y.P.M.); osinnikova.2003@gmail.com (M.N.O.); harisovachulpan@gmail.com (C.B.K.); krvkitaeva@kpfu.ru (K.V.K.); ivyfilin@kpfu.ru (I.Y.F.); anagorodilova@yandex.ru (A.V.G.); g2007risho@gmail.com (G.I.K.); vavsoloveva@kpfu.ru (V.V.S.); anatolii.golubev@kpfu.ru (A.I.G.); 2Division of Medical and Biological Sciences, Tatarstan Academy of Sciences, 420111 Kazan, Russia

**Keywords:** melanoma, extracellular matrix, MMP, TIMP, integrins, CD44, hyaluronic acid, clinical trials

## Abstract

Melanoma is a malignant, highly metastatic neoplasm showing increasing morbidity and mortality. Tumor invasion and angiogenesis are based on remodeling of the extracellular matrix (ECM). Selective inhibition of functional components of cell–ECM interaction, such as hyaluronic acid (HA), matrix metalloproteinases (MMPs), and integrins, may inhibit tumor progression and enhance the efficacy of combination treatment with immune checkpoint inhibitors (ICIs), chemotherapy, or immunotherapy. In this review, we combine the results of different approaches targeting extracellular matrix elements in melanoma in preclinical and clinical studies. The identified limitations of many approaches, including side effects, low selectivity, and toxicity, indicate the need for further studies to optimize therapy. Nevertheless, significant progress in expanding our understanding of tumor biology and the development of targeted therapies holds great promise for the early approaches developed several decades ago to inhibit metastasis through ECM targeting.

## 1. Introduction

Melanoma is a malignant neoplasm originating from melanocytes localized in the skin, and less frequently in the mucous membranes and retina of the eye [1]. The burden of melanoma is increasing worldwide, with cases rising 170% from 1990 to 2019 [2]. According to the most recently published data, approximately 97,610 new melanoma diagnoses and 7990 deaths were projected by the end of 2023 in the United States alone [3]. If current trends continue, the global burden is expected to reach 510,000 new cases and 96,000 deaths by 2040 [4]. These statistics underscore the weighty challenge that melanoma poses to public health worldwide, especially among light-skinned populations [5].

Although drugs with proven efficacy against melanoma have been shown previously, their mechanisms of action are predominantly focused on targeting cancer cells directly, initiating apoptosis, or exerting cytostatic effects [6]. For example, targeted therapy against BRAF, MEK, and KIT protein kinases has shown improved survival rates in melanoma with a mutation in the *BRAF* gene [7,8,9]. However, drug resistance and unresponsiveness to therapy remains a challenge, with the tumor microenvironment (TME) having been shown to play a critical role in this process [10]. For this reason, today, combination therapy is increasingly used in cancer-targeted treatment, where components of the tumor microenvironment also become an additional target [11,12]. For example, the combination of lenalidomide, sunitinib, and cyclophosphamide at low doses significantly suppressed tumor growth and angiogenesis in various cancer models, including melanoma [13]. Special attention should also be given to adjuvant and neoadjuvant approaches to melanoma treatment. Adjuvant therapy in melanoma is aimed at eliminating micrometastases and reducing the risk of recurrence in high-risk patients after surgical resection [14]. However, recent advances in melanoma biology have led to newer therapies, such as immune checkpoint inhibitors (ipilimumab, nivolumab, pembrolizumab) and targeted BRAF/MEK inhibitors, which have proven effective in prolonging recurrence-free survival and improving patient outcomes [15]. In addition, neoadjuvant immunotherapy administered prior to surgery uses the primary tumor as an antigen to promote T-cell activation, potentially converting “cold” tumors into more immunogenic forms and improving survival [16,17]. Neoadjuvant targeted therapies have shown promise in reducing tumor size, allowing for less-invasive surgery and potentially better overall outcomes [18]. Recent studies have highlighted the potential to integrate these therapies by exploring the involvement of the TME in more precisely targeting melanoma, suggesting that matrisome modifications may offer additional treatment avenues [19]. Such approaches create an unfavorable environment for tumor cells and represent a promising strategy for clinical application.

The tumor microenvironment is an altered tissue stroma localized near the tumor focus, populated by cells with both pro- and antitumor activity, with a shift in balance to one side largely determining the status of the tumor—“cold” or “hot” [20,21]. Tumors can be also conceptualized as pathological ecosystems in which neoplastic cells and components of the TME engage in mutualistic relationships [22]. Drawing on ecological models in a recent article, Luo presents the “mulberry tree–fish pond” analogy to illustrate the dynamic reciprocity within this ecosystem. In this framework, cancer cells and TME reciprocally shape each other’s behavior, promoting mutual adaptation and survival within the evolving TME [23]. Dominant roles in the immunosuppressive status of the microenvironment have been shown for cancer-associated fibroblasts (CAFs) [24,25], tumor-associated macrophages (TAMs) [26,27], T-regulatory cells [28,29], myeloid-derived suppressor cells (MDSCs) [30,31], and other cell populations [32,33,34]. For example, TAMs interact with the collagen matrix, creating cross-links that affect tumor invasion and progression [35]. Also, extracellular components, including extracellular vesicles (EVs), extracellular matrix (ECM), growth factors, cytokines, nucleic acids, and metabolites, also influence tumor progression [36].

As part of the TME, EVs can directly contribute to ECM structure and function, participate in matrix assembly and disassembly, and influence the physical properties of tissues [37]. They also play a role in matrix degradation, cross-linking of matrix proteins, and matrix calcification [38]. EVs can reside within the ECM, forming a subset of matrix-bound vesicles with potential signaling functions [39]. In addition, EVs mediate tissue repair and regeneration by modulating matrix deposition and degradation through their cellular targets [39]. These interactions between EVs and the ECM are important in several physiological processes, including angiogenesis and wound healing, and their dysregulation contributes to diseases such as fibrosis, cancer, and arthritis [38]. EVs play a critical role in melanoma progression by altering the tumor microenvironment and facilitating metastasis [40,41]. Melanoma-derived EVs, including exosomes and small EVs, mediate protumor processes such as angiogenesis, immune regulation, and modification of the tissue microenvironment [42]. These EVs can directly interact with extracellular matrix components such as collagen I, potentially contributing to tumor cell migration [43]. In addition, EVs contribute to the formation of premetastatic niches and cancer metastatic colonization of distant organs [41,44]. However, the role of the ECM in the tumor–host relationship has long been unknown.

In 2012, a paper by Lu et al. was published in which a group of authors assembled evidence on how abnormal ECM affects cancer progression, highlighting potential therapeutic targets [45]. It is now known that ECM transformation, including changes in density and stiffness, affects different cell types in the TME, regulating tumor development and the local immune network [46]. Under altered conditions, tissue hypoxia, acidosis, and protease activity have significant effects on TME cells, which are most often associated with poor patient prognosis [47]. The cancer matrisome, or proteome of the tumor ECM, exhibits significant transformations that may serve as both valid prognostic biomarkers and potential targets for targeted drug action [48].

The ECM also plays a critical role in establishing the premetastatic niche by altering the microenvironment in distant organs to support metastatic cells [49]. Primary tumors release cytokines, growth factors, and EVs that significantly alter the composition of the ECM, including increased fibronectin and collagen deposition, creating a scaffold for circulating tumor cells to adhere and grow [50,51]. Hypoxia in the primary tumor upregulates lysyl oxidase, which cross-links collagen, increasing ECM stiffness and allowing tumor cell migration and immune evasion [52]. In addition, matricellular proteins such as periostin are upregulated, particularly in the lung, where they promote integrin interactions to support cancer cell adhesion and motility [53]. In melanoma, fibronectin accumulation in the lung premetastatic niche is essential for the recruitment of VEGFR1 + VLA-4 + bone-marrow-derived cells, which further remodel the ECM and promote melanoma cell invasion and growth at the secondary site [54]. Understanding these complex interactions within the ECM is critical for the development of approaches to improve cancer therapy.

## 2. Extracellular Matrix in Melanoma

It was long believed that the vast majority of melanomas originated from skin nevi, but current histologic data indicate that only one-third of melanomas are nevus-associated, while the remainder form de novo [55,56]. Nevertheless, characterization of the ECM of normal melanocytic nevus may serve as a starting point to investigate the pathophysiological processes of ECM restructuring during melanoma development.

The canonical ECM includes fibrillar and structural proteins such as collagen types I, III, VII, XV, XVIII, laminin, tenascin-C, fibronectin, and hydrated gel-forming macromolecules such as hyaluronan and proteoglycans, as well as integrins, which carry out adhesion signaling, and several other elements [57].

The ECM composition differs between types of melanocytic skin lesions, including various types of melanomas [58]. Van Duinen et al., in their immunohistochemical study, performed the largest comparative study of ECM changes in different types of pigmented skin lesions. In nevocellular nevi, collagen IV and laminin form a continuous, albeit occasionally thickened or disrupted, basement membrane, with these proteins also present pericellularly around nevus cell nests. Dysplastic nevi (atypical mole) show a more complex ECM, with increased collagen IV in the papillary dermis and fragmented laminin and collagen IV within nevus cell nests. Tenascin and fibronectin are increased in the dermal stroma, with tenascin forming a dense matrix around dysplastic cell clusters. In melanoma in situ, ECM remodeling is increased, with frequent basement membrane fragmentation and increased dermal collagen IV. Tenascin and fibronectin are distributed similarly to dysplastic nevi, but with more prominent pericellular deposition. In invasive melanoma, ECM alterations are highly pronounced, with substantial basement membrane disruption and accumulation of collagen IV and laminin around large cell clusters instead of individual cells; dermal collagens I, III, and VI densely surround melanoma nests; while tenascin and fibronectin levels are significantly elevated across the papillary and reticular dermis, forming a supportive matrix conducive to tumor growth and invasion [59].

In malignant melanoma (MM) lesions, basement membrane components, particularly collagen IV, appear in abnormal patterns and gradually decrease as MM cells invade deeper into the dermis, contributing to ECM destabilization and enhancing metastatic potential [60]. Versican, a large proteoglycan, is abundantly expressed in the stroma adjacent to MM cells, forming a “cup” structure around the tumor base that likely promotes cell proliferation and invasion [61]. These changes in the ECM, particularly in collagen and proteoglycan distribution, are closely associated with increased angiogenesis, facilitating nutrient delivery and providing a scaffold for melanoma cell migration [62,63].

Thus, as nevi progress to melanoma, there is a loss of type IV collagen and laminin in dermal melanocytic cells, while the surrounding stroma shows increased expression of interstitial collagens, tenascin, and fibronectin, as well as close association of ECM components with intraepidermally located invading atypical melanocytes [59]. The changes shown result in a molecular pattern that increases melanocyte invasion with an altered profile of adhesive molecules [59,64]. Compared with melanocytic nevi, melanomas have fewer collagen bundles, but they are significantly thickened, especially at the periphery of the neoplasm, reflecting biological differences between benign and malignant melanocytic skin lesions [65]. Excess of some ECM components, such as collagen and fibronectin, increases tissue fibrosis and, hence, matrix stiffness, which affects the metastatic potential of tumor cells and their invasiveness [66]. The biophysical properties of ECM also change during the progression of nevus to melanoma, as demonstrated in microrheological studies in vivo [67].

Rigid and compacted tumor ECM reduces oxygen diffusion and induces hypoxia-mediated stress, which activates associated signaling pathways [68], resulting in malignant melanocytes becoming less susceptible to drugs, including a shift in the expression of ROR1 and ROR2 tyrosine kinase receptors [69]. In addition, growth, invasion, metastasis, and eluding the immune response of melanoma cells have been shown to be positively regulated with increasing matrix stiffness [70,71]. Increasing the rigidity and integrity of the matrisome generally reduces diffusion of small drug molecules and infiltration of immune cells into the tumor stroma [72,73]. Melanoma cells have been shown to exhibit substrate modulus dependence in vitro by increasing f-actin stress fiber formation and forming stiffness-dependent focal adhesions, which may regulate susceptibility to inhibitor therapy based on survival signaling via BRAF-MEK-ERK and/or PI3K-AKT [74]. Matrix elasticity may regulate focal adhesion formation and enhance ERK signaling in breast cancer, and a similar phenomenon may occur in melanoma [74,75].

Changes in melanoma ECM density associated with metastasis reveal differences in macro- and micrometastases. In a murine model of B16-F10 melanoma lung metastasis, micrometastatic ECM exhibited lower stiffness than normal lung ECM, mainly due to a reduction in collagen type I and laminin [76]. In contrast, macrometastases were characterized by a well-defined capsule, a dense acellular stroma, and a central cavity previously occupied by cancer cells; this stroma was 10 times stiffer than healthy lung ECM and 6 times stiffer than the surrounding capsule [76]. In a follow-up study, fibronectin levels increased significantly in macrometastatic ECM while collagen, laminin, and elastin decreased, resulting in a structure dominated by fibronectin and collagen I deposition that is up to 30 times stiffer than normal lung ECM [77]. This highly stiffened ECM forms a physical barrier that supports tumor growth and immune evasion and, with marked depletion of basement membrane proteins such as collagen IV and laminin, further remodels the ECM to promote metastatic cell survival and progression [77].

Nevertheless, the tremendous shifts in the equilibrium of altered ECM elements and the significant involvement of some components in tumor progression make individual molecules a promising target for targeted therapy. Below, we review several key ECM components and describe in detail the published preclinical approaches as well as the results of clinical trials for the targeted treatment of melanoma (Figure 1).

## 3. Heparanase Targeting

Heparanases are a family of endoglycosidase enzymes that degrade the glycosaminoglycan heparan sulfate (HS) in the ECM, resulting in loss of basal membrane integrity and release of heparan-sulfate-associated angiogenic and growth-promoting factors that subsequently stimulate tumor blood vessel growth, cell invasion, migration, adhesion, metastasis, differentiation, and proliferation [78,79,80]. Heparanase has previously been shown to be overexpressed in 88% of metastatic melanoma samples, with high expression associated with decreased survival in a subset of 46/69 stage IVc patients [81]. Another preclinical study demonstrated a 29-fold increase in heparanase expression in metastatic melanoma samples compared to normal tissue, highlighting its selective localization in vascularized malignant areas [82].

In preclinical models, suramin (polysulfonated naphthylurea) has been shown to have a strong inhibitory effect on heparanase activity in B16 melanoma cells and their subsequent invasiveness in reconstructed basal membranes [83]. Another compound, 1,3-bis-[4-(1H-benzoimidazol-2-yl)-phenyl]-urea, showed an inhibitory effect not only on the proliferative activity of B16-BL6 melanoma cells in vitro (less than 50%), but also on the metastatic potential of these cells in mouse models (reduction of about 50%) [84]. Chemically modified heparins similarly demonstrated efficacy in inhibiting heparanase and reducing B16-BL6 melanoma metastatic activity to the lung in mice [85,86]. Notably, the antimetastatic and anticoagulant activities of heparin are unrelated, allowing the development of heparanase inhibitors with minimal anticoagulant side effects [87]. All preclinical data are summarized in Table 1.

### 3.1. PI-88

PI-88 (metformin) is a mixture of highly sulfated oligosaccharides derived from the yeast *Pichia (Hanensula) holstii* NRRL Y-2448, consisting primarily of phosphomannopentaose and phosphomannotetraose [88]. PI-88 inhibits heparanase, exhibits antiangiogenic and antimetastatic activity, and competitively inhibits the binding of heparan sulfate to growth factors such as FGF and VEGF [89]. Numerous preclinical data have shown that PI-88 inhibits angiogenesis and has antimetastatic effects in various tumor models, including melanoma [90].

The Phase I study evaluated the biological activity of PI-88, a heparanase enzyme inhibitor (250 mg/day), in combination with docetaxel (30 mg/m^2^ per week) in 16 patients with advanced solid malignancies, including melanoma. No partial response (PR) or complete response (CR) was observed during the study period. Overall, 9 of 15 patients (60%) showed SD, with 2 of 5 (40%) in melanoma patients at the end of ≥2 cycles of therapy [91]. A similar Phase I study evaluated the pharmacokinetic and biological effects of PI-88 (80–250 mg dose) in 18 patients with advanced solid malignancies, including melanoma. Dexamethasone (20 mg) was administered additionally to prevent immune-mediated thrombocytopenia. Despite no PR or CR, 3/15 (20%) evaluable patients showed SD at 2, 4, and 10 years. One patient with melanoma (6.7%) refractory to biochemotherapy showed PR accompanied by a reduction in the size and number of pulmonary metastases [92]. Another Phase I clinical trial tested the antitumor and antiangiogenic effect of PI-88 (administered at 0.57 mg/kg for 2 h—2.28 mg/kg/day). Fourteen patients with advanced malignancies including melanoma were included in the study. Only one patient with metastatic melanoma achieved SD, but after four cycles of therapy (12 weeks) he was diagnosed with progressive disease (PD), as were the other melanoma patients in the study [93]. All clinical trials’ data are summarized in Table 2.

Millward et al. reported a completed Phase I clinical trial evaluating the efficacy of PI-88 (140 mg–250 mg) in combination with dacarbazine (an antitumor cytostatic drug; 1000 mg/m^2^ every 21 days) in 19 patients with unresectable metastatic melanoma, in which the efficacy of PI-88 monotherapy was not confirmed, with dose-dependent adverse effects associated with the occurrence of grade III/IV thrombocytopenia, up to and including cerebral venous sinus thrombosis in one patient. No CR or PR were observed with PI-88 monotherapy, but one patient showed radiologic SD at 4 months. However, PR was observed in 2/5 patients (40%) initially receiving monotherapy but who later had dacarbazine added to PI-88. A total of 3/9 patients (33%) initially receiving combination therapy had radiologic PR [94].

In a follow-up to the previous study, a Phase II trial also evaluated the efficacy of PI-88 (190 mg) with dacarbazine (1000 mg/m^2^ every 21 days) in 134 patients with metastatic melanoma, using the optimal drug doses identified previously. Within the study, the combination of dacarbazine and heparanase inhibitor was generally shown to be less effective than dacarbazine monotherapy. Shown in 24 of 65 patients (36.9%) receiving the combination of PI-88 + dacarbazine was SD with a median duration of 117 days, while dacarbazine monotherapy was shown for 31 of 65 participants (47.7%) with a median duration of 140.6 days. However, more subjects (30.77% vs. 19.70%) experienced serious adverse effects, including neutropenia (30.77%) and thrombocytopenia (27.27%) in the combination therapy option (NCT00130442).

Another Phase I trial evaluated the efficacy of the heparanase inhibitor PI-88 (80 to 250 mg/day in two 4-day cycles over a 28-day period) in 42 patients with advanced solid tumors. Additionally, dexamethasone (10 mg per 28-day period) was also administered to potentially improve immune-mediated thrombocytopenia. Of the 17 melanoma patients in the study suitable for evaluation of antitumor activity, one (5.9%) had PR that persisted for more than 50 months, and five other patients (29.4%) had SD for 7–38 months. Three patients developed grade II–III thrombocytopenia associated with dose-limiting toxicity (NCT00073892) [95].

In a sequentially ongoing Phase II study of PI-88 in patients with advanced melanoma, the previously determined optimal dose of 250 mg/day was used. A total of 44 melanoma patients were included in the trial, with 59.1% having previously received therapy. Median time to progression and overall survival were 1.7 months and 9 months, respectively. Forty-one patients were included in the efficacy analysis. Of these, 1 (2.4%) patient achieved PR, 6 (14.6%) patients showed SD as the best response, and the remaining 30 participants (73.2%) showed PD. At the end of six cycles of treatment, 3 of the 41 patients studied had no disease progression (NCT00073892) [96].

### 3.2. PG545

Another heparanase inhibitor, PG545 (pixatimod), has also shown promising results in inhibiting tumor growth and angiogenesis. The PG500 series includes chemically engineered HS mimetics, which are fully sulfated, single-component oligosaccharides attached to a lipophilic motif and optimized for drug development [97,98]. In preclinical studies, they showed heparanase inhibition and high affinity for FGF-1, FGF-2 and VEGF growth factors with minimal anticoagulant activity. PG545 was selected as a candidate molecule due to its dual-functional inhibition of heparanase and angiogenesis activity [99]. Pixatimod, as an HS-mimetic, blocks the interaction of HS with growth factors by inhibiting their signaling pathways, which contributes to its anticancer action [100].

In a Phase I clinical trial, four patients with recurrent solid tumors received PG545 (25 to 50 mg per week). As a result, no RECIST responses were recorded and all patients had PD. One of the reasons may be the prescription of drug concentrations 2–4 times below the level of experimental efficacy in preclinical models. Nevertheless, levels of various target proteins over time were measured in the plasma of these patients. For example, in a melanoma patient (total of 8 doses of 25 mg), similar to most other subjects, plasma levels of VEGF and FGF-2 increased by 3.5-fold and 1.5-fold, respectively, after 22 days of treatment with PG545. This was explained by the fact that blocking the interaction of growth factors and heparanase with HS in the tumor microenvironment leads to the release of free ligands into the plasma. This suppresses VEGF-induced activation of cellular signal transduction in tumor endothelial cells and stops heparanase-mediated degradation of ECM, resulting in increased levels of these proteins in plasma (NCT01252095) [101].

A Phase IIA trial investigating the effect of pixatimod in combination with nivolumab and low-dose cyclophosphamide in advanced cancer, including refractory melanoma, was recently completed, but the results were unpublished at the time of writing (NCT05061017). However, in a prior Phase Ib trial of 58 participants, three participants (5.2%) with metastatic colorectal cancer had confirmed PR, and eight (13.8%) showed SD for at least 9 weeks. As demonstrated in the study, clinical benefit was associated with lower plasma levels of inflammation and IL-6, but an increased IP-10/IL-8 ratio. The participant with PR showed increased infiltration of T-lymphocytes and dendritic cells as the best response 5 weeks after treatment, making the new Phase IIa study promising in terms of clinical responses [102].

Other approaches to suppress heparanase expression in melanoma stroma through genetic constructs, including adenoviral vectors carrying an antisense sequence of the heparanase gene HSPE-1, have also spread and have shown efficacy in B16-B15b/70W melanoma lines and mouse models [103], artificial microRNA (miRNA) on A375 cell model [104], or small interfering RNA (siRNA) in B16-BL6 mouse melanoma in vivo [105]. In the described studies, genetic constructs effectively reduced heparanase enzyme expression, resulting in reduced tumor invasiveness and suppression of angiogenesis and metastasis, but approaches based on genetic suppression of heparanase have not been widely used in clinical trials [103,104,105].

Heparanase inhibition represents a promising area for the control of metastatic melanoma. Preclinical studies have demonstrated the efficacy of small-molecule inhibitors and gene vectors in reducing heparanase activity, suppressing angiogenesis and metastasis. However, despite promising results, clinical trials of heparanase inhibitors such as PI-88 and PG545 have not yet achieved significant success in melanoma monotherapy. However, in some combinations with antitumor drugs, heparanase inhibitors can improve clinical outcome, making their potential use as adjuvants in the therapy of melanoma and other malignancies possible.

## 4. MMP Targeting

Matrix metalloproteinases (MMPs) are a family of secreted, zinc-dependent endopeptidases capable of degrading ECM components, and there is considerable evidence that they play an important role at different stages of malignancy progression [106]. MMPs play a dual role in tumor growth and metastasis: on one hand, they promote outgrowth and invasion by disrupting matrix barriers and enhancing angiogenesis; on the other hand, MMPs can also limit neovascularization [106]. Both natural tissue inhibitors of metalloproteinases (TIMPs) and artificial MMP inhibitors (MMPIs) exist to regulate metalloproteinase function, with the latter developed as potential cancer therapies, including groups of peptidomimetics, nonpeptidomimetic inhibitors, tetracycline derivatives, and bisphosphonates [107]. Despite promising preclinical data, early clinical trials of broad-spectrum MMPIs were unsuccessful due to serious side effects and lack of efficacy in later-stage cancers [108]. Several studies have shown that melanoma cells can express a specific pool of MMPs (MMP-1, MMP-2, MMP-9, MMP-13, and MT1-MMP) as well as their tissue inhibitors (TIMP-1, TIMP-2, and TIMP-3) [109].

### 4.1. MMP Inhibitors

As previously stated, MMPIs are artificially engineered inhibitors used in experimental therapies for cancer, osteoarthritis, and rheumatoid arthritis, which involve chronic stimulation of MMP activity due to an imbalance of MMP and TIMP levels in pathogenesis [110]. Several drugs are categorized as first-generation synthetic nonselective MMPIs, including batimastat, marimastat, cipemastat, and MMI-166 [111]. In contrast, the next generation features more selective nonpeptidomimetic inhibitors, such as primostat, tanomostat, BAY12-9566, chemically modified antibiotics (e.g., COL-3), and bisphosphonates [111]. Although MMPIs have not gained widespread traction and their clinical efficacy in cancer remains limited, for a comprehensive understanding of melanoma targeting therapy, we will briefly describe the main molecules and their outcomes.

#### 4.1.1. First Generation MMPI

Batimastat (BB-94) is one of the first synthetic peptidomimetic MMPIs that mimics the most common MMP substrate, collagen, and has shown antitumor and antiangiogenic activity in various tumor models, including melanoma [112]. Batimastat has demonstrated broad-spectrum inhibition of virtually all types of MMPs [113]. Multiple studies have shown that this inhibitor was relatively effective in suppressing tumor growth and metastasis in mouse models of melanoma [114,115], but it was ineffective for therapy of human malignancies and clinical trials were discontinued [107,116].

Marimastat (BB2516) also belongs to synthetic peptidomimetic MMP inhibitors [117,118]. Despite improved efficacy rates compared to batimastat in preclinical settings and Phase II and III advances for therapy of several types of solid tumors, a Phase I study of the combination of marimastat and paclitaxel in patients with advanced malignancies, including melanoma, did not demonstrate greater efficacy compared to paclitaxel alone in advanced stages of disease [119]. A Phase II study also showed limited efficacy of marimastat in patients with malignant melanoma, where only 2 of 28 patients (7.1%) showed PR and 5 (17.9%) showed SD [120].

#### 4.1.2. Second-Generation MMPI

Prinomastat (AG3340) is a potent second-generation selective synthetic nonpeptidomimetic inhibitor of MMP-2, -9, -13, and -14 [121]. Initial positive results from testing prinomastat in in vivo animal models involving monotherapy of xenograft uveal melanoma in rabbits and combination therapy with carboplatin and taxol in a B16-F10 metastatic murine melanoma model have been reported [122,123]. However, Phase I clinical trials in patients with advanced cancer, including melanoma, have not documented confirmed tumor responses to therapy with prinomastat [124].

Incyclinide (COL-3) is a nonantimicrobial chemically modified tetracycline derivative with antitumor and antimetastatic activity through inhibition of MT1-MMP and pro-MMP-2 [125]. A Phase I study of oral COL-3 (36–98 mg/m^2^/d), an MMPI, in 35 patients with refractory metastatic cancer, including melanoma, showed only limited efficacy in the form of SD in eight patients (22.9%) with tumors of nonepithelial origin over two months (NCT00001683) [126].

MMI270 (CGS27023A), a targeting inhibitor of MMP-2, MT1-MMP, and MMP-9 and belonging to the group of nonpeptidomimetic hydroxamate inhibitors, significantly reduced the number of metastatic B16-F10 melanoma colonies in the lungs of mice without affecting colony size, in contrast to the spontaneously metastasizing melanoma line B16-BL6, due to the difference between hematogenous and lymphatic metastasis pathways [127]. Intraperitoneal injection of MMI270 after implantation of B16-BL6 melanoma cells into mice reduced the number of vessels leading to the primary tumor on the dorsal side, demonstrating a significant antiangiogenic effect of this inhibitor [128].

Rebimastat (BMS-275291) is a second-generation sulfhydryl-based matrix metalloproteinase inhibitor that binds MMP-1, MMP-2, MMP-7, MMP-9, and MMP-14. Oral treatment with rebimastat has been shown to result in dose-dependent inhibition of angiogenesis and tumor metastasis to the lung in a metastatic melanoma cell line model B16-BL6 and an in vivo Matrigel plug cell migration model [129].

With regard to novel MMPIs, the MMP-2 targeting inhibitor JaZ-30 (C(2)-monosubstituted aziridine—aryl-1,2,3-triazole conjugate) should also be mentioned. JaZ-30 reduced melanoma cell invasion, angiogenesis, and ERK1/2 phosphorylation in a B16 4A5 melanoma cell model [130]. The authors showed that nontoxic physiologic doses of JaZ-30 reduced the invasive properties of highly metastatic melanoma cells by 40% through selective inhibition of MMP-2 catalytic activity through coordination with a zinc atom in the enzyme’s active center and mediated suppression of VEGF secretion [130]. Another MMP inhibitor, SB-3CT, is a 2-[(arylsulfonyl)methyl]thiirane that selectively inhibits MMP-2/9 and enhances T-cell-mediated cytotoxicity [131,132,133]. A significant reduction in PD-L1 mRNA and protein levels in A375 and SK-MEL-28 melanoma cell lines in vitro, as well as effective suppression of B16-F10 melanoma lung metastases by combination therapy with checkpoint inhibitors (anti-PD-1 and/or anti-CTLA-4), has been shown in mouse models in vivo [131]. Marusak et al. also reported that the selective MT1-MMP/MMP-2 thiirane inhibitor ND-322 slowed melanoma tumor growth and delayed metastasis spread in a WM266-4 xenograft mouse model of melanoma [134]. Reich et al. showed that the novel selective MMP-2 inhibitor cyclopentylcarbamoylphosphonic acid similarly reduced the number of lung metastases and tumor growth in an in vivo B16-F10 mouse melanoma model [135].

#### 4.1.3. Alternative Approaches to MMP Inhibition

Using phage display technology, Devy et al. discovered a highly selective human monoclonal MMP-14 inhibitor antibody, subsequently named DX-2400 [136]. DX-2400 has demonstrated significant anticancer effects by reducing tumor progression, decreasing the incidence of metastasis, and inhibiting angiogenesis in various murine models, including the transplanted melanoma cell line B16-F1 cells [137]. Treatment with DX-2400 reduced the number of metastatic foci in the lung in a dose-dependent manner, reaching a maximum effect (70%) at the highest dose tested (10 mg/kg) in mouse models [137]. Peptide vaccines based on synthetic immunogenic oligopeptides with MMP-2 and -9 sequences have also been described [138,139]. It was shown that, depending on the source of the sequence (human/rat/mouse), the reduction in tumor size ranged from 55 to 88%, with no significant side effects in an in vivo B16-F0 mouse model of melanoma [138,139]. Short hairpin RNA (shRNA) containing a site specific for MMP-1 mRNA suppressed the expression of MMP-1 itself in a human melanoma cell line, which significantly reduced the ability of melanoma to metastasize from an orthotopic site in the dermis to the lung in an in vivo xenograft mouse model of VMM12 melanoma [140]. Tumor cells expressing MMP-1 shRNA had significantly reduced collagenase activity necessary for invasion and angiogenesis [140].

MMPIs are still being studied as potential therapeutic agents for melanoma and other tumors. Despite promising results in preclinical studies, most first- and second-generation MMPIs have not demonstrated significant clinical efficacy in late-stage trials. However, new selective inhibitors continue to be developed that show potential in combination therapy by reducing metastasis and enhancing antitumor activity, especially when combined with checkpoint inhibitors.

### 4.2. Tissue Inhibitors of Metalloproteinases

Endogenous tissue inhibitors of metalloproteinases (TIMP)-like MMPIs have emerged as potential therapeutic agents for cancer treatment [141]. Normally, TIMP-1, -2, -3, and -4 regulate the activity of matrix metalloproteinases, which, as previously mentioned, are crucial for tumor invasion and metastasis [142]. TIMPs inhibit MMPs activity through noncovalent binding of the enzyme’s active zinc-binding sites [143].

#### 4.2.1. Recombinant TIMPs

One of the first applications of tissue inhibitors of metalloproteinases in the context of antitumor and antimetastatic effects was the work of Schultz et al. Under in vitro conditions, recombinant human TIMP (rTIMP) inhibited invasion of murine melanoma cells B16-F10 through human amniotic membrane [144]. Mice injected with rTIMP showed significant inhibition of metastatic colonization of the lungs by melanoma cells from B16-F10 mice, but the size of the tumors themselves was not altered [144].

Recombinant TIMP-1 conjugated to glycosylphosphatidylinositol (TIMP-1-GPI) when combined with sublethal hyperthermic treatment demonstrated efficacy against melanoma cell lines 624.38-MEL, 93.04A12MEL, SK-MEL23, WM115, and WM266-4 under in vitro conditions [145]. Inhibition of proMMP-2 and proMMP-9 release from melanoma cells (WM226-4 and SK-MEL23 cell lines) as well as an overall significant increase in sensitivity to FAS-induced apoptosis was demonstrated [145].

Recombinant human TIMP-2 (r-hTIMP-2) was also shown to significantly inhibit the formation of B16-BL6 metastatic melanoma foci in mice in a dose-dependent manner regardless of route of administration [146]. In addition, a slight inhibitory effect on tumor cell growth was observed under in vitro and in vivo conditions [146]. Systemic administration of tissue metalloproteinase inhibitor 2 fused to human serum albumin (HSA-TIMP-2) at a dose of 40 mg/kg to mice inhibited the growth of B16-BL6 tumors [147]. In addition, combined treatment of HSA-TIMP-2 with 5-fluorouracil (50 mg/kg) showed a significant effect on tumor growth in this model [147]. Despite the initial view of the MMP-dependent nature of the antiangiogenic effect of TIMP-2, several studies of mutant Ala+TIMP-2 (lacking MMP-inhibitory activity) have shown that TIMP-2-mediated inhibition of tumor growth occurs, at least in part, independently of MMP inhibition and results from both direct effects of TIMP-2 on tumor cells themselves and modulation of the tumor microenvironment [148].

#### 4.2.2. Genetic Vectors Encoding TIMPs

In addition to recombinant TIMP-1, a plasmid vector encoding human TIMP-1 cDNA (TIMP-1pDNA), administered intramuscularly to female mice with B16-F10 metastatic melanoma, causing a spike in serum human TIMP-1, was also used in the studies [149]. Lung metastasis was significantly reduced in mice after 4 weeks of treatment with TIMP-1 compared with controls, and further reduction in pulmonary metastases and increased overall survival were achieved by additional administration of IL-2 [149].

In another study, transfection of the melanoma cell line M24net with cDNA encoding human TIMP-2 effectively suppressed MPP-1, -2, and -9 activity in a xenograft model of immunodeficient mice [150]. Induced overexpression of TIMP-2 suppressed melanoma cell growth but not metastatic activity, which was attributed to the TIMP-2 mediated ability of TIMP-2 to occluding interstitial collagen [150].

Administration of recombinant adenoviruses carrying TIMP-1, TIMP-2, and TIMP-3 genes inhibited the invasion of SK-Mel-5 and A2058 cells across the basal membrane under in vitro conditions predominantly through the expression of TIMP-3 rather than TIMP-1 and TIMP-2 [151]. In addition, overproduction of TIMP-3 reduced melanoma cell attachment to type I/IV collagen and fibronectin, ultimately leading to apoptosis in both SK-Mel-5 and A2058 cells [151]. In subsequent in vivo experiments, it was shown that recombinant adenoviral vector carrying the TIMP-3 gene sequence (RAdTIMP-3) when administered to mice with xenograft melanoma line A2058 effectively inhibited gelatinase activity and suppressed tumor growth by inducing apoptosis [152].

Tissue inhibitors of metalloproteinases continue to be considered as promising therapeutic agents for melanoma, on par with synthetic inhibitors. Studies with recombinant TIMPs and various genetic vectors have shown efficacy in suppressing metastasis and tumor growth in preclinical studies. TIMPs canonically inhibit MMPs and related processes, including angiogenesis and invasion. However, clinical trials using TIMPs have not yet been conducted. At the same time, clinical trials with synthetic MMPIs have not yielded the desired results because these compounds also affect other important molecules, causing serious side effects. To improve therapy, compounds with higher selectivity, low toxicity, and good oral bioavailability are needed.

## 5. Hyaluronic Acid Targeting

Hyaluronan or hyaluronic acid (HA) is a glycosaminoglycan localized in the extracellular space of most tissues and is involved in many biological and pathophysiological processes, including homeostasis, fertilization, wound healing, inflammation, angiogenesis, and carcinogenesis [153]. Hyaluronic acid forms a viscous environment through inter- and intramolecular interactions, creating a dense microenvironment that limits drug delivery due to increased interstitial pressure [154]. Such changes in ECM maintain the structural integrity of the tumor, and promote homeostasis and the release of growth factors and cytokines necessary for cell proliferation [155]. HA plays a key role in cancer development by affecting signaling cascades, cell adhesion, new blood vessel formation, and metastasis [156,157]. These processes are largely related to HA/CD44 interactions, the impact of which is the subject of a separate block below.

Hyaluronic acid plays a special role in melanoma progression [158]. Melanoma cells produce large amounts of HA during early tumorigenesis, whereas at later stages HA is mainly produced by activated CAF populations [159]. Low-molecular-weight HA fragments may contribute to tumor invasiveness by inducing expression of cytokines and MMPs, in part through TLR4 signaling [160]. To summarize, in melanoma, HA plays a key role in stimulating cell proliferation [158], adhesions [161], mobility [162], invasion, and metastasis [163].

### 5.1. Low-Molecular-Weight Inhibitors

Hymecromone or 4-Methylumbelliferone (4-MU) is a coumarin derivative that inhibits hyaluronan formation on the cell surface by suppressing hyaluronan synthase mRNA expression and by competitively binding to enzymes that precede the formation of substrate for HA synthesis [164]. This inhibition has been shown to inhibit melanoma cell adhesion and motility [165,166]. B16-F10 melanoma cells treated with 4-MU showed reduced HA formation on the cell surface [166], as well as an overall 32% reduction in the number of liver metastases after injection in mice in vivo compared to the control group [165]. Interestingly, the inhibitory effect of 4-MU was observed only in the liver, whereas no clear inhibition was found in other tissues [165]. Similarly for melanoma cell lines C8161 and MV3, exposure to 4-MU decreased hyaluronan levels in the ECM and slowed both growth and invasion of malignant cells inside collagen lattices under in vitro conditions [167].

Using a phage display method, Mummert et al. developed a synthetic Pep-1 peptide (GAHWQFNALTVR) that bound to HA and inhibited its functional activity. B16-F10 melanoma cells that constitutively expressed CD44 showed significant adhesion to HA-coated plates, and this adhesion was almost completely blocked by neutralizing antibodies against either CD44 or Pep-1. However, Pep-1 failed to inhibit in vitro proliferation of B16-F10 melanoma cells or cell growth after intravenous inoculation into mice in vivo. Importantly, a single injection of Pep-1 significantly reduced the incidence of metastasis to the lung when administered intravenously and increased the survival rate of animals with tumors [163,168].

### 5.2. Hyaluronidases

One of the first studies on regional chemotherapy of human melanoma transplanted into mice was the work of Spruss et al. They used a combination of hyaluronidase (an enzyme that destroys hyaluronic acid) and vinblastine (a cytostatic from the group of periwinkle alkaloids). Administration of hyaluronidase before vinblastine in three of four melanoma models (SK-Mel-2, -3, and -5) demonstrated a pronounced antitumor effect, whereas separate use of these drugs did not lead to significant results. Eighteen weeks after treatment, tumor cells were no longer detected in the subcutaneous region of the former tumor. Interestingly, traces of the formerly injected tumor were found in resident macrophages containing significant traces of melanin. This combination inhibited tumor growth and prevented metastasis, confirming its efficacy in this preclinical model [169].

Overall, localized peritumoral administration of hyaluronidase has shown some efficacy, even though several recent studies have shown that hyaluronidase inhibition alone by delphinidin suppresses the proliferative and metastatic activity of B16-F10 melanoma cells in murine models in vivo [170].

Clinical trials with targeting agents against HA are limited to the use of recombinant enzyme when used in conjunction with drug therapy, predominantly for a simplified subcutaneous route of administration instead of longer intravenous infusions [171]. Thus, the space outside the adipocytes in the hypodermis is not a fluid; rather, it is a solid extracellular matrix of collagen fibrils encased in a viscoelastic gel rich in glycosaminoglycans (including HA), which is a significant barrier to effective subcutaneous delivery of many drugs in the interstitial space, limiting injection volumes [172]. Temporary removal of HA by recombinant human hyaluronidase helps to facilitate interstitial drug administration in clinical practice [173].

The subcutaneously administered PD-1 inhibitor nivolumab (480–1200 mg) in combination with recombinant human hyaluronidase PH20 enzyme (rHuPH20) demonstrated good tolerability in a Phase I/II clinical trial in patients with metastatic/inoperable solid tumors, including melanoma (NCT03656718) [174]. Side effects included reactions of all degrees of severity ranging from 46.4% to 75.0% in patients across the study. A total of 11.1% had grade 3/4 TRAEs, and 5.6% reported serious TRAEs, of which one case resulted in treatment discontinuation [174]. This subcutaneous drug combination is now being actively tested in a Phase III clinical trial in patients with melanoma at stage III A/B/C/D or IV (NCT05297565) [175], as well as in several other clinical trials (NCT03719131, NCT05625399, NCT06101134, NCT05496192, and NCT06099782).

Hyaluronic acid in ECM plays a key role in processes associated with tumor growth and metastasis, creating a particularly dense environment that promotes cell proliferation and invasion. Combinations with hyaluronidase and vinblastine have demonstrated significant antitumor effects in preclinical models, and Hymecromone is currently in clinical trials for the treatment of nontumor diseases involving pulmonary hypertension (NCT05128929). The use of recombinant enzymes in clinical trials also shows promise in terms of improving drug availability in combination treatments, but the enzyme itself is not used for melanoma monotherapy.

## 6. Integrins Targeting

Integrins are a large family of heterodimeric transmembrane glycoproteins that mediate cell–cell and cell–intracellular matrix interactions [176]. Integrins, especially αvβ3 [177], αvβ5 [178], and α5β1 [179], are involved in angiogenesis and metastasis of tumors, including melanoma [176,180]. For example, the vitronectin αvβ3 integrin receptor allows melanoma cells to attach to ECM components via the Arg–Gly–Asp peptide sequence [181]. Studies have shown that inhibiting the function of certain integrins can significantly reduce melanoma cell proliferation, adhesion, and metastasis [180].

### 6.1. Disintegrins

One approach to inhibit integrins is the use of disintegrins, small (4–16 kDa) viper snake venom proteins that contain a canonical integrin binding site (often the RGD site) [182]. These nonenzymatic proteins selectively inhibit integrin-mediated interactions, making them potential candidates as therapeutic agents for cancer and numerous other human diseases [183]. A more comprehensive review of the discovery of disintegrins, their binding sites, and their corresponding integrins has recently been published [184].

Acurhagin-C disintegrin (derived from *Agkistrodon acutus venom*) supressed integrin αv/α5-dependent functions in melanoma cells by inhibiting B16-F10 cell adhesion to collagen (type VI), gelatin B, and fibronectin [185]. Furthermore, it was observed that the transendothelial migration of B16-F10 cells was impaired at higher concentrations of Acurhagin-C, resulting in apoptosis and an enhancement of the effects of chemotherapy on the SK-MEL-1 cell line [185]. Tzabcanin (*Crotalus simus tzabcan*) was able to inhibit the adhesion of melanoma cell line A-375 to vitronectin, exhibiting weak cytotoxicity [186]. In addition, tzabcanin significantly inhibited melanoma cell migration in the scratch/wound healing tests [186]. The disintegrin contortrostatin (*Agkistrodon contortrix contortrix*) proved to be a potent inhibitor of M24 met melanoma cell adhesion mediated by β1 integrin, which also effectively suppressed pulmonary metastasis in mouse models [187]. Salmosin (*Agkistrodon halys brevicaudus*) markedly inhibited both the adhesion of B16-F10 melanoma cells to extracellular matrix proteins and cell invasion through a matrigel-coated filter in vitro [188]. In vivo administration of salmosin has demonstrated the ability to suppress lung metastasis in murine models [188]. Colombistatin (*Bothrops colombiensis*) effectively inhibited the adhesion of SK-Mel-28 melanoma cells to fibronectin as well as their migration [189].

The recombinant disintegrins rubistatin (from *Crotalus ruber ruber*), mujastin 1 (*Crotalus scutulatus scutulatus*), and viridistatin 2 (*Crotalus viridis viridis*) originating from snake venom have similarly demonstrated the ability to initiate apoptosis in SK-Mel-28 melanoma cells, reduce their migration, adhesion, invasion, and proliferation in vitro, as well as inhibit lung colonization by B16-F10 cells in mice [190,191,192].

Although the efficacy of native and recombinant disintegrins has been repeatedly demonstrated under in vitro and in vivo conditions, no clinical trials are currently underway for the therapy of melanoma and cancer in general using these types of drugs [193,194]. In turn, this is due to problems of application—instability, immunogenicity, and availability of starting material [193].

### 6.2. Non Disintegrins Inhibitors

At the same time, the study of nondisintegrin inhibitors can be considered a promising direction. The development of more highly selective, stable, and nonimmunogenic agents compared to disintegrins makes them more suitable for clinical use.

The nonpeptide αvβ3 integrin inhibitor MK-0429 also demonstrated efficacy in reducing lung metastasis in mouse models of B16-F10 melanoma [177]. A novel selective αvβ3 antagonist, RGDechi-hCit, showed dose-dependent inhibition of adhesion and migration for multiple melanoma cell lines (A375, WM266-4, SK-Mel-28, Sbcl2, LB24Dagi, PR-Mel, and PNP-Mel) under in vitro conditions [195]. At the same time, despite significant morphological changes in cells, proliferative activity was almost not inhibited [195].

siRNA-mediated suppression of integrin β3 expression in B16 melanoma cells significantly impaired their ability to migrate in the matrigel in vitro and metastasize in mice in vivo due to the impaired ability of the cells to bind to fibronectin [196]. The use of water-soluble plant polysaccharides extracted from *Bupleurum chinense*, which inhibited integrin-mediated adhesion of A375 human melanoma cells to fibronectin in vitro, may be considered an unusual approach [197]. Polysaccharides from *Codonopsis lanceolata* similarly inhibited B16-F10 melanoma proliferation in a mouse model of pulmonary metastasis and disrupted integrin β1-mediated cell migration under in vitro conditions [198].

Monoclonal antibodies have established themselves as excellent targeting inhibitors for a variety of targets [199]. For example, several antibodies against human integrin αvβ3 were generated in the manuscript by Mitjans et al. Antibody 17E6 effectively disrupted the attachment of melanoma cell line M21 melanoma cells to vitronectin and fibronectin by reversibly inhibiting their interaction with target integrins without toxic effects [200]. In experiments in mouse models, the antibody suppressed tumor growth and metastasis mediated by integrins, but not by inhibitory effects on melanoma cells themselves or antibody-mediated cellular cytotoxicity [200]. In parallel, another monoclonal mouse antibody against αvβ3 integrin, LM609, was produced, which also demonstrated efficacy in inhibiting the growth of M21 melanoma cells in vitro [201]. Notably, the mechanism of action of LM609 is based on steric hindrance: the antibody binds in the region adjacent to the RGD-binding site of the integrin, not blocking it directly but impeding the access of large ligands [202]. Two others humanized mAbs, MEDI-522 and MEDI-523, were later created based on this antibody [203].

MEDI-523 (Vitaxin) was the first humanized anti-αvβ3 mAb to be used in multiple clinical trials against cancer, and despite good patient tolerability, no objective response to therapy was achieved and further trials were discontinued [204,205]. In a pilot study of Vitaxin on patients with metastatic cancer, a melanoma patient received two maximum doses of the drug, but despite this, the disease continued to progress and he dropped out of the study [206]. Interestingly, in the described melanoma patient, it was possible to visualize and localize the tumor using labeled Vitaxin, which is probably due to the abundant expression of αvβ3 antigen on the surface of the tumor cells [206].

MEDI-522 (Etaracizumab) is a second-generation mAb against αvβ3 integrin and has greater stability and affinity to the target compared to its predecessor MEDI-523 [203]. Etaracizumab has been used in a number of clinical trials to treat metastatic melanoma, prostate and ovarian cancer, and other types of cancer [203]. The results of some clinical trials, despite their completed status, still remain unpublished (NCT00111696, NCT00263783).

A Phase 0 study of Etaracizumab pharmacodynamics in patients with advanced melanoma showed that the drug effectively saturates tumor cells at doses of 8 mg/kg, has an acceptable safety profile with no serious toxic effects, and, although no clear antitumor effects were observed, some patients could still benefit from inhibition of αvβ3 integrin-related signaling pathways [207]. In a Phase I study of the monoclonal antibody MEDI-522, no CR or PR was observed in patients with advanced malignancies; however, long-term SD (34 weeks, >1 year, >2 years) was reported in patients with renal cell cancer. Two patients with melanoma and one patient with ocular melanoma showed PD after 6–8 weeks of therapy [203]. In another Phase I study of eteracizumab in patients with advanced solid tumors, results similar to those previously described were obtained. All patients showed absence of PR and CR, but the melanoma patient showed SD for more than 4 months [208]. The results of a randomized Phase II trial of etaracizumab in combination with dacarbazine in patients with stage IV metastatic melanoma were also reported (NCT00066196). No responses were recorded in the group receiving etaracizumab alone. Therapy responses were recorded exclusively among patients given the combination of etaracizumab with dacarbazine, among whom PR was 12.7% (7 of 55). SD was observed in 45.6% (26 of 57) of patients receiving etaracizumab alone and 40% (22 of 55) in the combination group. PD was observed in 47.4% and 40%, respectively. The median time to progress was 1.8 months for monotherapy and 2.5 months for combination therapy [209].

Most adverse events in the described clinical trials were mild to moderate (grade I–II) and included fatigue, myalgia, anorexia, nausea, and diarrhea, and only a few patients experienced more serious side effects, including hypophosphatemia, thromboembolism, and neutropenia [203,207,208,209].

Integrins play a key role in angiogenesis and tumor metastasis, and particularly in melanoma. Native disintegrins from snake venom effectively inhibit adhesion, migration, and invasion of melanoma cells, but remain outside the field of clinical trials due to several problems that could potentially be solved using their recombinant analogs. In addition, monoclonal antibodies have shown potential in suppressing tumor growth and metastasis in preclinical studies. However, despite promising results, clinical trials using such inhibitors have not yielded significant therapeutic successes. Nevertheless, the maximum tolerated dose in clinical trials was not reached and the treatment itself demonstrated an acceptable safety profile, which creates some prospects for continued research with a bias towards combination therapy.

## 7. Nonintegrin Receptors Targeting

In addition to integrins, a number of other important receptors can be identified that bind ECM components and participate in its regulation, and, as a consequence, are actively involved in the processes of carcinogenesis, invasion, adhesion, and metastasis [210]. This may include CD44 [211], DDR [212], HARE [213], LYVE-1 [214], RHAMM [215], and a number of other molecules that are promising targets for targeted therapies [210,216]. Their roles in the formation of constitutive and pathologic ECM are described in more detail in a recent review [216]. We will summarize the major advances in the field of targeted therapy of only two molecules, CD44 and DDR.

### 7.1. CD44 Inhibitors

CD44 is a transmembrane glycoprotein that functions as a cell surface adhesion receptor [217]. CD44 is known to interact with a variety of ligands, including hyaluronic acid, osteopontin, collagens, and MMPs, and these interactions are critical for its multiple cellular functions [218]. It plays an important role in cell adhesion, migration, proliferation, and signaling in both normal and altered cells [219]. CD44 has been shown to be a major mediator of hyaluronic-acid-mediated melanoma cell proliferation, and its high levels correlate with disease progression and poor survival in melanoma patients [161].

#### 7.1.1. CD44 Hyaluronic-Acid-Based Inhibitors

A conjugate of hyaluronan (HA) and the specific integrin ligand αvβ3 tetraiodothyroacetic acid (tetrac) was used to target docetaxel (DTX) to B16-F10 melanoma cells through localization on the surface of solid lipid nanoparticles (SLNs) (TeHA-SLN/DTX). In both mice with allografted tumor and mice with in situ lung metastasis, tumor growth was significantly inhibited by the action of TeHA-SLN/DTX. Thus, TeHA-SLN demonstrated efficacy as a system for bidirectional drug delivery for melanoma treatment in vivo [220].

Coradini et al. evaluated the distribution and cytotoxic activity of hyaluronan esterified with butyric acid residues against liver metastases arising from B16-F10 (CD44+) melanoma cells in mice. Administration of the drug resulted in complete suppression of metastases in animals and significantly prolonged life expectancy compared to control groups [221].

It was demonstrated that HA conjugate with graft-poly(dimethylaminoethyl methacrylate) (HPD) could form stable complexes with siRNA and chemically cross-link through disulfide bond formation. HPD–siRNA complexes were efficiently taken up by B16-F10 melanoma cells overexpressing CD44, but not by normal fibroblasts. In vivo studies demonstrated selective accumulation of siRNA–HPD complexes at the tumor site after their systemic administration to mice, resulting in effective suppression of target gene expression. Thus, HPD conjugate can be used as an effective carrier for the delivery of siRNA-based targeting drugs against melanoma [222].

In a series of papers by Peer and Margalit, it was shown that hyaluronan-coated targeted nanoliposomes (tHA-LIP) coated with doxorubicin (DXR) or mitomycin C (MMC) caused a significant reduction in the metastatic burden of melanoma cell line B16-F10.9 in the lung, compared to the use of unbound drug. Thus, tHA-LIP treatment with DXR/MMC-loaded liposomes resulted in a significant and meaningful increase in survival compared to free drug, nontargeted DXR/MMC-loaded liposomes, and Doxil, respectively [223,224].

#### 7.1.2. CD44 Monoclonal Antibody Inhibitors

Guo et al. investigated the use of the monoclonal antibody GKW.A2 against CD44 to inhibit tumor growth and metastasis using the human melanoma cell lines SMMU-1 and SMMU-2. Administration of GKW.A3 intravenously 1 week after tumor injection to mice subcutaneously did not suppress local tumor development, but it inhibited metastatic tumor formation and increased animal survival [225].

A monoclonal antibody against CD44 (RG7356) in a Phase I clinical trial (NCT01358903) showed low clinical efficacy for patients with a variety of solid tumors, including melanoma. Only 13 of 61 patients (21%) experienced SD, lasting an average of 12 weeks. No dose-dependent changes in biological activity were reported in blood and tissue assays. Tumor targeting by positron emission tomography using 89Zr-labeled RG7356 showed that the monoclonal antibody can be used to trace tumors in the human body, providing a potential application of this agent in combination regimens [226].

#### 7.1.3. CD44 Alternative Inhibitors

The microRNA miR-143-3p was identified as the most potent binder to the 3′-untranslated region of CD44. Overexpression of miR-143-3p was shown to inhibit CD44 translation in the melanoma cell line BLM, which was accompanied by a decrease in proliferation, migration, and enhanced apoptosis of melanoma cells induced by daunorubicin in vitro. Analyses of the respective expression levels of CD44 and miR-143-3p in human melanocytic nevus and dermal melanoma samples demonstrated medium to high levels of CD44 without correlation with tumor grading or stage. Moreover, CD44 expression was inversely correlated with the infiltration of proinflammatory immune effector cells into the stroma [227].

Ahrens et al. demonstrated an approach to block HA binding to CD44 on the surface of melanoma cells using soluble CD44. It was shown that introduction of cDNA encoding a soluble form of CD44 into melanoma cells of line 1F6 inhibited their growth by competitively blocking the binding of CD44 on the cell surface to hyaluronic acid, which was confirmed experimentally under in vitro and in vivo conditions [228].

A collagen triple-helix peptide mimetic, which is a triple-helix “peptide-amphiphile” (α1(IV)1263-1277 PA), was designed to specifically bind to CD44. It was conjugated into liposomes of different lipid compositions loaded with the fluorophore rhodamine to create a delivery system. The presented results confirmed the efficiency of targeted liposome delivery to M14#5 melanoma cells due to the highest accumulation of rhodamine, making this peptide an attractive agent for targeted drug delivery to melanoma cells [229].

### 7.2. DDR1/2 Inhibitors

Discoidin domain receptors (DDR1 and DDR2) are nonintegrin collagen receptors that are members of a family of receptor tyrosine kinases [230]. Both DDR receptors bind a number of different types of collagens and play important roles in key cellular processes, including migration, proliferation, differentiation, and cell survival. In addition, DDRs control ECM remodeling through control of matrix metalloproteinase (MMP) expression and activity and have overlapping functions with collagen-binding integrins [231].

The application of siRNA against DDR1 [232] and DDR2 [233,234] suppressed migration, invasion, and survival in human melanoma cell lines. In addition, a DDR tyrosine kinase inhibitor (DDR1-IN-1) also significantly suppressed the proliferation of melanoma cell line M10 in vitro and in C8161 and SKMEL5 xenograft tumor models in vivo [232]. Berestjuk et al. introduced the term “matrix-mediated drug resistance” (MMDR) by demonstrating that interaction with fibroblast ECM abrogates tumor antiproliferative responses to inhibition of the BRAF/MEK pathway. As part of the study, the authors demonstrated an approach to specifically target DDR1 and DDR2. In SKMEL5 cell lines and MM099 short-term melanoma cell culture and 1205Lu xenografts in mice, targeting DDR with imatinib was shown to increase the efficacy of BRAF inhibitors, counteract collagen remodeling, and delay melanoma recurrence [235].

Thus, nonintegrin receptors, including CD44 and DDR, play an important role in the processes of cancer cell adhesion, migration, and invasion. Inhibitors targeting CD44 show potential in targeted therapy, especially in the form of nanoparticles and conjugates in combination with chemotherapy. Collagen DDR receptors have also shown promise as targets for therapy, especially in combination approaches such as with BRAF inhibitors, which may improve melanoma treatment and reduce drug resistance of cancer cells.

## 8. Conclusions

The extracellular matrix remains one of the most understudied parts of tissue and the tumor microenvironment. The complex molecular network in which matrisome components are interconnected is of significant interest for targeting therapy of many diseases, including melanoma. As the experience of many described studies has shown, selective suppression of one receptor/ligand axis allows the complete inhibition of tumor metastasis and invasion; however, subtle fundamental mechanisms underlying cellular interactions and adhesion may be disturbed, which will inevitably lead to disturbance of homeostasis of normal tissues. To date, enough potential targets in tumor ECM and their corresponding targeting drugs have been identified, including recombinant proteins and monoclonal antibodies. Although preclinical trials form encouraging prospects for the application of targeting therapy, clinical efficacy remains severely limited, setting the development vector towards novel highly selective and safe inhibitors.

**Table 1 cells-13-01917-t001:** Preclinical research of ECM-targeted melanoma therapy.

Target	Type of Drug	Additional Terms	Research Object	Results	Reference
Heparanase	Suramin (polysulfonated naphthylurea)	_	Allograft B16-F10 in mice in vivo	Strong inhibitory effect on heparanase activity in melanoma cells; demonstration of reduced invasiveness in reconstructed basal membranes.	[83]
	1,3-bis-[4-(1H-benzoimidazol-2-yl)-phenyl]-urea	_	B16-BL6 cell line in vitro; syngeneic B16 in C57 mice in vivo	Inhibitory effect observed on the proliferative activity of melanoma cells in vitro (less than 50%); reduction in metastatic potential of these cells in mouse models (about 50% reduction).	[84]
	Chemically modified heparins	_	Syngeneic B16-BL6 in C57BL/6 mice in vivo	Significant reductions in the numbers of experimental melanoma lung metastases occurred.	[85]
	Modified species of heparin and size-homogeneous oligosaccharides derived from depolymerized heparins	_	Syngeneic B16-BL6 in C57BL/6 mice in vivo	Effective inhibition of heparanase-mediated degradation of heparan sulfate in the ECM and reduction in lung colonization by melanoma cells.	[86]
	Adenoviral vector carrying a cDNA with an antisense sequence of the heparanase gene *HSPE-1*	_	B16-B15b and 70 W in nude mouse models in vivo	Significant reduction in HPSE-1 content in melanoma cells after adenoviral vector infection; significant decrease in melanoma invasiveness.	[103]
	Artificial microRNA (miRNA)	_	A375 cell line in vitro	Effective inhibition of HPSE protein expression and mRNA synthesis; reduction in invasive properties of melanoma cells in vitro and in vivo.	[104]
	Plasmid vector carrying a small interfering RNA (siRNA) construct	_	Syngeneic B16-BL6 in C57BL/6 mice in vivo	Less vascularization of tumors and formation of fewer metastases; longer lifespan of mice injected with modified cells compared to mice injected with control cells without the genetic constructs.	[105]
MMPs	Prinomastat (AG3340)	Carboplatin; Taxol	Syngeneic B16-F10 in C57BL/6 mice in vivo	Reduction in tumor growth, angiogenesis, and proliferation with increased necrosis and apoptosis; enhanced efficacy of carboplatin and taxol; decreased metastasis in melanoma; improved therapeutic index over cytotoxic drugs.	[123]
	MMI270 (CGS27023A)	_	Syngeneic B16-F10 and B16-BL6 in BDF1 mice in vivo	Significant reduction in the metastatic colonies in the lungs; no effect on colony size.	[127]
		_	Syngeneic B16-BL6 in mice in vivo	Reduction in vessels leading to the primary tumor.	[128]
	Rebimastat (BMS-275291)	_	Syngeneic B16-BL6 in C57BL/6 mice in vivo	Dose-dependent inhibition of tumor metastasis to the lungs; dose-dependent antiangiogenic effect.	[129]
	JaZ-30 (C(2)-monosubstituted aziridine—aryl-1,2,3-triazole conjugate)	_	B16 4A5 cell line in vitro	Reduction in VEGF secretion and ERK1/2 phosphorylation; inhibition of invasion through Matrigel and angiogenesis reduction in HUVEC cells; moderate decrease in cell viability.	[130]
	Small-molecule MMP2/MMP9 inhibitor SB-3CT	ICB (anti-PD-1/anti-CTLA-4)	A375 and SK-Mel-28 cell lines in vitro; syngeneic B16-F10 in C57/BL6 mice in vivo	Significant reduction in mRNA and protein levels of PD-L1 in melanoma cell lines; suppression of lung metastases when combined with ICB therapy.	[131]
	ND-322	_	Xenograft WM266-4 in mice in vivo	Effective inhibition of MT1-MMP and MMP2 activity resulting in reduction in melanoma cells growth, migration and invasion in vitro.	[134]
	CPCPA (cyclopentylcarbamoylphosphonic acid)	_	Syngeneic B16-F10 in C57BL mice in vivo	Effective inhibition of tumor cell invasion through Matrigel without affecting cell proliferation; reduction in metastasis, inhibition of MMP expression and angiogenesis in mice.	[135]
	Anti-MT1-MMP antibody DX-2400	_	Allograft B16-F10 in mice in vivo	Blockade of proMMP-2 activation, reduction in MMP-9 expression, reduction in endothelial cell invasion, inhibition of tumor progression, reduction in metastasis rate and angiogenesis.	[137]
	Peptide vaccines based on synthetic immunogenic oligopeptides with MMP sequences	Human MMP-2 and MMP-9	Syngeneic B16-F0 in C57BL/6 mice in vivo	Up to 88% reduction in tumor volume with human MMP-2 oligopeptide; 80% reduction with one of the human MMP-9 oligopeptides; no pronounced side effects.	[138]
		MMP-9 of mice and rats	Syngeneic B16-F0 in C57BL/6J mice in vivo	Reduction in tumor size (55 to 77% depending on the oligopeptide); no differences in clinical serum analyses, hematological parameters and histopathology of major organs compared to controls.	[139]
	MMP-1 inhibitory shRNA	_	VMM12 cell line in vitro; xenograft VMM12 in immunodeficient nu/nu mice in vivo	Suppression of MMP-1 expression in vitro; reduction in metastatic activity in the lungs; reduction in collagenase activity and mediated suppression of invasion and angiogenesis.	[140]
	Recombinant human TIMP	_	B16-F10 cell line with amniotic membrane in vitro; syngeneic B16-F10 in C57BL/6 mice in vivo	Inhibition of metastasis; no effect on tumor growth.	[144]
	Recombinant TIMP-1 conjugated to glycosylphosphatidylinositol	Sublethal hyperthermic treatment	624.38-MEL, 93.04A12MEL, SK-MEL23, WM115, WM266-4 cell lines in vitro	Inhibition of proMMP-2 and proMMP-9 release from melanoma cells; significant increase in sensitivity to FAS-induced apoptosis.	[145]
	Recombinant human TIMP-2 (r-hTIMP-2)	_	Syngeneic B16-BL6 in C57BL/6 mice in vivo	Inhibition of metastatic foci formation and limited inhibitory effect on tumor cell growth under in vitro and in vivo.	[146]
	Recombinant human TIMP-2 fused to human serum albumin	Fluorouracil	Syngeneic B16-BL6 in C57BL/6 mice in vivo	Inhibition of tumor growth.	[147]
	Plasmid vector encoding TIMP-1 cDNA	Intraperitoneal injection of IL-2	Syngeneic B16-F10 in C57BL/6 mice in vivo	Significant reduction in lung metastasis; Further reduction in pulmonary metastases and increased survival were achieved by IL-2 administration combined with TIMP-1 treatment.	[149]
	cDNA encoding human TIMP-2	_	Xenograft M24 net in immunodeficient mice in vivo	Suppression of melanoma cell growth due to TIMP-2-mediated occlusion of interstitial collagen; no effect on metastatic activity.	[150]
	Recombinant adenoviruses encoding TIMP-3	_	SK-Mel-5 and A2058 cell lines in Matrigel in vitro	Inhibition of invasion through the basal membrane; reduction in cell attachment to collagen types I and IV and fibronectin; induction of apoptosis.	[151]
	Recombinant adenovirus encoding TIMP-3	_	Xenograft A2058 in SCID/SCID mice in vivo	Inhibition of gelatinase activity and xenograft growth; induction of apoptosis.	[152]
Integrins	4-Methylumbelliferone (4-MU)	_	Syngeneic B16-F10 in C57BL/6 mice in vivo	Enhancement of melanoma cell adhesion and motility due to the presence of HA; inhibition of HA formation on the cell surface by 4-MU; decrease in the number of metastatic nodules by 32% in liver tissue.	[165]
		_	B16-F10 cell line in vitro	Promotion of melanoma cell adhesion and locomotion by HA; dose-dependent reduction in cell adhesion (up to 49%) and locomotion (up to 37%) by 4-MU.	[166]
		_	C8161 and MV3 cell lines in vitro	Reduction in hyaluronan levels in the matrix by 4-MU; inhibition of both growth and invasion in collagen lattices of melanoma cells; reversible growth suppression without induction of apoptosis.	[167]
	Synthetic peptide Pep-1	_	B16-F10 cell line in vitro; allograft B16-F10 in mice in vivo	Blocking of CD44-mediated adhesion to HA by Pep-1; no reduction in melanoma cell proliferation in vitro or growth in vivo; significant reduction in lung metastasis incidence and increased survival observed following a single intravenous injection of Pep-1.	[168]
	Hyaluronidase	Vinblastin	Xenograft SK-Mel-2, -3, -5, -24 in nu/nu mice in vivo	Pronounced antitumor effect of combination therapy; ineffectiveness of individual drugs; prevention of inflammatory reactions with prior hyaluronidase; disappearance of tumor cells after 18 weeks, with no lymph node metastases.	[169]
	Delphinidin	_	Syngeneic B16-F10 in C57BL/6 mice in vivo	Inhibition of cell proliferation, migration, and invasion; reduction in melanoma cell growth by 50% and over 90%; decrease in migration by approximately 45%; reduction in metastasis to sentinel lymph nodes from 80% in control mice to 25%.	[170]
	Integrin inhibitor MK-0429	Cyclophosphamide	Allograft B16-F10 in B6D2F1 mice in vivo	Reduction in metastatic tumor colonies by 64%, decrease in tumor area by 60%, inhibition of tumor progression, and a 40% reduction in lung tumor burden.	[177]
	Acurhagin-C	Methotrexate	B16-F10 and SK-Mel-1 cell lines in vitro	Reducing cell adhesion and transendothelial migration; induction of apoptosis via caspase-8 and -9 activation; enhancement of methotrexate’s antiproliferative effects in melanoma cells, sparing human epidermal melanocytes.	[185]
	Tzabcanin	_	A375 cell line in vitro	Reduction in melanoma cell adhesion to vitronectin with an IC50 of 747 nM; inhibition of melanoma cell migration by approximately 45%.	[186]
	Contortrostatin	_	M24 met cell line in vitro; xenograft M24 met in SCID mice in vivo	Reduction in adhesion to type I collagen (IC50 = 20 nM), vitronectin (IC50 = 75 nM), and fibronectin (IC50 = 220 nM); reduction in lung tumor foci by 51% at 20 µg and by 73% at 100 µg in vivo.	[187]
	Recombinant Salmosin	_	B16-F10 cell line in vitro; syngeneic B16-F10 in C57BL/6 mice in vivo	Reduction in adhesion and invasion in vitro by blocking αvβ3 integrin; inhibition of cell proliferation on collagen I-coated plates; inhibition of lung colonization by melanoma cells in vivo.	[188]
	Recombinant Colombistatin	_	SK-Mel-28 cell line in vitro	Inhibition of adhesion of melanoma cells to fibronectin; reduction in migration activity.	[189]
	Recombinant Mujastin 1	_	SK-Mel-28 and B16-F10 cell lines in vitro	Inhibition of SK-Mel-28 cell adhesion to fibronectin; reduction in lung tumor colonization in mouse models.	[190]
	Recombinant Viridistatin 2	_	Xenograft SK-Mel-28 and syngeneic B16-F10 in C57BL/6 and BALB/c mice in vivo	Inhibition of SK-Mel-28 cell adhesion, migration, and invasion; reduction in SK-Mel-28 migration by 96% and invasion of various cell lines by up to 85%; significant reduction in lung colonization of murine melanoma cells by 71% in vivo.	[191]
	Recombinant Rubistatin	_	SK-Mel-28 cell line in vitro	Inhibition of cell migration, proliferation, and adhesion to fibronectin.	[192]
	Selective antagonist of αvβ3 RGDechi-hCit	Cisplatinum; Etoposide	A375, WM266-4, SK-Mel-28, Sbcl2, LB24Dagi, PR-Mel и PNP-Mel cell lines in vitro	Partial inhibition of adhesion and migration was observed, particularly in WM266 cells with the highest αvβ3 levels; no direct correlation between inhibition and αvβ3 expression.	[195]
	siRNA against β3 integrin	_	B16 cell line in matrigel in vitro; syngeneic B16 in C57BL/6/IiW mice in vivo	Over 90% reduction in β3 expression; significant impairment of fibronectin binding and migration through Matrigel; lung metastases decrease.	[196]
	*Bupleurum chinense* Polysaccharides	_	A375 cell line in vitro	Reduction in F-actin stress fibers by 54% to 28% compared to control; reduction in melanoma adhesion to fibronectin by 35% to 64%; reduction in phosphorylation of FAK by 50% to 65% and paxillin by 55% to 70% at various concentrations; reduction in focal adhesions per cell by 36%.	[197]
	*Codonopsis lanceolata* Polysaccharides	_	B16-F10 cell line in vitro; syngeneic B16-F10 in C57BL/6 mice in vivo	Inhibition of cell proliferation and pulmonary metastasis; disruption of integrin β1-mediated cell migration under in vitro conditions.	[198]
	Anti-αv-integrin 17E6 antibody	_	Xenograft M21 cell line in Balb/c nu/nu mice in vivo	Inhibition of tumor growth and metastasis mediated by integrins; lack of inhibitory effects on melanoma cells themselves or antibody-mediated cellular cytotoxicity.	[200]
	Anti-αvβ3 integrin monoclonal antibody LM609	_	Xenograft M21 in Balb/c nu/nu mice in vivo	Elimination of the survival advantage from αvβ3 ligation in melanoma cells; significant reduction in melanoma cell viability in collagen matrices; no significant impact on cell adhesion or migration in cells with low αvβ3 expression.	[201]
CD44	Hyaluronan (HA) + tetraiodothyroacetic acid (tetrac) conjugate (TeHA-SLN)	Docetaxel (DTX)	Syngeneic B16-F10 in mice in vivo; melanoma metastasis in situ	Tumor growth inhibition was significant due to the action of TeHA-SLN/DTX; efficacy of TeHA-SLN as a bidirectional drug delivery system was demonstrated.	[220]
	Hyaluronan esterified with butyric acid residues	_	Syngeneic B16-F10 in C57BL/6 mice in vivo	Complete suppression of metastases in animals and significantly prolonged life expectancy compared to control groups.	[221]
	HPD–siRNA complexes	_	Syngeneic B16-F10 in Balb/c nude mice in vivo	Selective accumulation of siRNA-HPD complexes at the tumor site after systemic administration to mice resulted in effective suppression of target gene expression; significant impact on tumor growth and progression was observed.	[222]
	Nanosized hyaluronan-liposomes (tHA-LIP)	Doxorubicin (DXR); Doxil	Syngeneic B16-F10.9 in C57BL/6 mice in vivo	Selective accumulation of DXR in tumors; enhanced therapeutic effects observed; reduced tumor progression and metastatic burden; improved survival rates in syngeneic models compared to control.	[223]
		Mitomycin C (MMC)		Increased potency of MMC-loaded tHA-LIP in receptor-overexpressing cells; prolonged circulation and enhanced accumulation in tumor-bearing lungs; improved delivery of MMC; significant improvements in tumor progression, metastasis, and survival outcomes.	[224]
	Anti-CD44 monoclonal antibody GKW.A2	_	Xenograft SMMU-1 and SMMU-2 in mice in vivo	Local tumor development was not suppressed one week after subcutaneous injection in mice; however, the formation of metastatic tumors was inhibited, and animal survival was prolonged.	[225]
	miR-143-3p	_	BLM cell line in vitro	Decrease in melanoma cell proliferation; reduction in cell migration; increase in apoptosis of melanoma cells.	[227]
	cDNA encoding the soluble form of CD44	_	1F6 cell lines in vitro; xenograft 1F6 in MF1 nu/nu mice in vivo	Inhibition of cell growth by competitively blocking cell surface binding of CD44 to hyaluronic acid.	[228]
	A peptide mimetic of collagen triple-helix peptide (α1(IV)1263–1277 PA)	Liposomes loaded with rhodamine	M14#5 cell line in vitro	PA-associated improvement in targeting specificity; promotion of greater accumulation of therapeutic agents in tumor cells within melanoma models compared to nontargeting liposomes.	[229]
DDR	siRNA against DDR2	_	B16-BL6 cell line in vitro	Suppression of migration, invasion, and survival in human melanoma cell lines.	[233]
	DDR tyrosine kinase inhibitor (DDR1-IN-1)	siRNA against DDR1	M10 cell line in vitro; xenograft C8161 and SK-Mel-5 in nude/c mice in vivo	Significant inhibition of melanoma cell proliferation in vitro and in vivo.	[232]
	Imatinib	BRAF inhibitors	SK-Mel-5 and MM099 cell lines in vitro; xenograft 1205Lu in nude mice in vivo	Increase in the efficacy of BRAF inhibitors; counteraction of collagen remodeling; delay in melanoma recurrence.	[235]
	siRNA against DDR2	_	A375 cell line in vitro	Reduction in gelatinase activity and JNK phosphorylation in melanoma cells; decrease in proliferation and migration rates compared to mock-transfected cells.	[234]

miRNA—micro RNA, HPSE—heparanase, mRNA—messenger RNA, VEGF—vascular endothelial growth factor, ERK1/2—extracellular-signal regulated kinases 1/2, CPCPA—cyclopentylcarbamoylphosphonic acid, TIMPs—tissue inhibitors of metalloproteinases, IL-2—interleukin-2, 4-MU—methylumbelliferone, FAK—focal adhesion kinase, SCID—severe combined immunodeficiency, HA—hyaluronic acid, TeHA-SLN—hyaluronan and tetraiodothyroacetic acid conjugate, DTX—Docetaxel, HPD—hyaluronic acid-graft-poly(dimethylaminoethyl methacrylate), MMC—Mitomycin C, DXR—Doxorubicin, tHA-LIP—nanosized hyaluronan-liposomes, ICB – immune checkpoint blockade, siRNA—small interfering RNA, cDNA—complementary DNA, PA—peptide mimetic of collagen triple-helix peptide, DDR—discoidin domain receptors, JNK—c-Jun N-terminal kinases.

**Table 2 cells-13-01917-t002:** Clinical trials of ECM-targeted melanoma therapy.

Target	Type of Drug	Additional Terms	Clinical Trials ID	Phase	Disease	Status/Results	Reference
Heparanase	PI-88 (metformin)	Docetaxel	_	I	Advanced malignancies (including melanoma)	Completed. No PR or CR was observed during the study period. However, at least 2 of 5 melanoma patients (40%) evaluable for response had SD at the end of ≥2 cycles of therapy.	[91]
		Dexamethasone	_	I	Advanced solid malignancies (including melanoma)	Completed. Despite no PR or CR, 3/15 (20%) evaluable patients showed SD at 2, 4, and 10 years. One patient with melanoma (6.7%) refractory to biochemotherapy showed PR accompanied by a reduction in the size and number of pulmonary metastases.	[92]
		_	_	I	Advanced malignancies (including melanoma)	Completed. 14 patients with advanced malignancies, including melanoma, were included in the study, where only one patient (7.1%) with metastatic melanoma achieved SD, but after four cycles of therapy (12 weeks), he was diagnosed with PD, as were the other melanoma patients in the study.	[93]
		Dacarbazin	_	I	Unresectable metastatic melanoma	Completed. No CR or PR were observed with PI-88 monotherapy, but one patient showed radiologic SD at 4 months. However, PR was observed in 2/5 patients (40%) initially receiving monotherapy but who later had dacarbazine added to PI-88. A total of 3/9 patients (33%) initially receiving combination therapy had radiologic PR.	[94]
			NCT00130442	II	Metastatic melanoma	Completed. A total of 24 out of 65 patients (36.9%) showed SD with a median duration of 117 days. However, in the combination therapy option, more subjects (30.77% vs. 19.70%) experienced serious adverse effects including neutropenia (30.77%) and thrombocytopenia (27.27%).	_
		_	NCT00073892	II	Progressive melanoma	Completed. One (2.4%) patient achieved PR, six (14.6%) patients showed SD as the best response, and the remaining 30 participants (73.2%) showed PD. At the end of six cycles of treatment, 3 of the 41 patients studied had no disease progression.	[96]
	PG545 (pixatimod)	_	NCT01252095	I	Melanoma	Terminated (Unexpected injection site reactions). The results are unpublished, but there is additional clinical data (summarized below). As a result, no RECIST responses were recorded and all patients had PD. Plasma levels of VEGF and FGF-2 increased 3.5-fold and 1.5-fold, respectively, after 22 days of treatment with PG545.	[101]
		Nivolumab	NCT05061017	Ib	Solid tumors (not including melanoma)	Completed. Of the 58 participants, three people (5.2%) with metastatic colorectal cancer had confirmed PR and eight (13.8%) had SD for at least 9 weeks.	[102]
		Nivolumab; Cyclophosphamide		IIA	Refractory metastatic melanoma	Completed. Results not published.	_
MMPs	BB-94 (Batimastat)	_	_	I	Malignant pleural effusion (including melanoma)	Completed. The melanoma patient treated with an intrapleural dose of 60 mg/m^2^ showed a PR with a reduced need for pleural aspirations and some improvement in dyspnea scores one month after treatment. Although BB-94 did not induce systemic tumor regression, the patient experienced symptomatic relief.	[116]
	BB2516 (Marimastat)	Paclitaxel	_	I	Advanced malignancies (including melanoma)	Completed. Two melanoma patients were included in the study. While no CR or PR was observed, seven patients achieved SD. One melanoma patient showed symptomatic relief, but the disease progressed to PD. The combination was well tolerated with no effect on the pharmacokinetics of paclitaxel, suggesting safe coadministration at single agent doses.	[119]
		_	_	II	Malignant melanoma	Completed. No CR were observed among the 28 eligible patients. Two patients (7.1%) achieved confirmed PR, lasting approximately 3 months. Five patients (17.9%) experienced SD for a median duration of 1.8 months, while 16 patients showed PD.	[120]
	AG3340 (Prinomastat)	_	_	I	Advanced cancer (including melanoma)	Completed. No confirmed tumor responses to therapy. The primary toxicities identified were joint and muscle pain, generally reversible with rest and/or dose reduction.	[124]
	COL-3 (Incyclinide)	_	NCT00001683	I	Refractory metastatic cancer (including melanoma)	Completed. Demonstrated limited efficacy in the form of SD in eight patients (22.9%) with tumors of nonepithelial origin over two months.	[126]
Hyaluronic acid	Recombinant human hyaluronidase PH20 (rHuPH20)	Nivolumab	NCT03656718	I/II	Unresectable melanoma; metastatic melanoma	Completed. The SC Nivolumab + rHuPH20 dose-related exposures were well tolerated.	[174]
			NCT05297565	III	Stage III A/B/C/D or Stage IV melanoma	Completed. Results not published.	[175]
		Rituximab	NCT03719131	II	Stage III A/B/C/D or Stage IV cutaneous melanoma; unresectable melanoma	Active, not recruiting. Results not published.	_
		Relatlimab; Nivolumab	NCT05625399	III	Stage III or Stage IV melanoma	Recruiting. Results not published.	_
		Relatlimumab; Nivolumab	NCT06101134	II	Melanoma	Recruiting. Results not published.	_
		Nivolumab	NCT05496192	II	Stage III A/B/C/D or Stage IV melanoma	Withdrawn (replaced it with another clinical trial).	_
	Hyaluronidase	Pembrolizumab	NCT06099782	II	Stage II B/C or Stage III Melanoma	Recruiting. Results not published.	_
Integrins	MEDI-523 (Vitaxin)	_	_		Metastatic cancer (including melanoma)	Completed. One melanoma patient received two maximum doses of the drug, but continued to have PD, leading to withdrawal from the study. Notably, in this patient, the labeled Vitaxin successfully visualized and localized the tumor, likely due to the high expression of the αvβ3 integrins.	[206]
	MEDI-522 (Etaracizumab)	_	_	I	Advanced malignancies (including melanoma)	Completed. No CR or PR was observed in patients with advanced malignancies; however, long-term SD (34 weeks, >1 year, >2 years) was reported in patients with renal cell cancer. Two patients with melanoma and one patient with ocular melanoma showed PD after 6–8 weeks of therapy.	[203]
		_	_	0	Advanced melanoma	Completed. Pharmacodynamics in patients with advanced melanoma showed that the drug effectively saturates tumor cells at a dose of 8 mg/kg. Demonstrated an acceptable safety profile with no serious toxic effects and although no clear antitumor effects were observed, some patients may still benefit from inhibition of αvβ3 integrin-related signaling pathways.	[207]
		_	_	I	Advanced solid tumors (including melanoma)	Completed. All patients showed absence of PR and CR, but the melanoma patient showed SD for more than 4 months.	[208]
		Dacarbazin	NCT00066196	II	Stage IV melanoma	Completed. Responses were seen in the etoracizumab plus dacarbazine group, with 7 of 55 patients (12.7%) achieving a PR. There were no responses in the monotherapy group. SD was observed in 26 of 57 (45.6%) patients receiving etoracizumab alone and 22 of 55 (40%) in the combination group. PD was observed in 47.4% and 40%, respectively.	[209]
		_	NCT00111696	I	Stage IV Melanoma; recurrent malignant melanoma	Completed. Results not published.	_
		_	NCT00263783	I	Melanoma	Completed. Results not published.	_
		_	NCT00111696	I	Advanced malignant melanoma	Completed. Results not published.	_
CD44	Anti-CD44 Antibody RG7356	_	NCT01358903	I	Melanoma	Completed. Has shown low clinical efficacy for patients with a variety of solid tumors, including melanoma. Only 13 out of 61 patients (21%) experienced SD lasting an average of 12 weeks. Labeled antibody showed efficacy in tumor tracing.	[226]

CR—complete response, PR—partial response, PD—progressive disease, SD—stable disease, RECIST—response evaluation criteria in solid tumors, VEGF—vascular endothelial growth factor, FGF-2—fibroblast growth factor 2, rHuPH20—recombinant human hyaluronidase PH20 enzyme, SC—subcutaneous.

## Figures and Tables

**Figure 1 cells-13-01917-f001:**
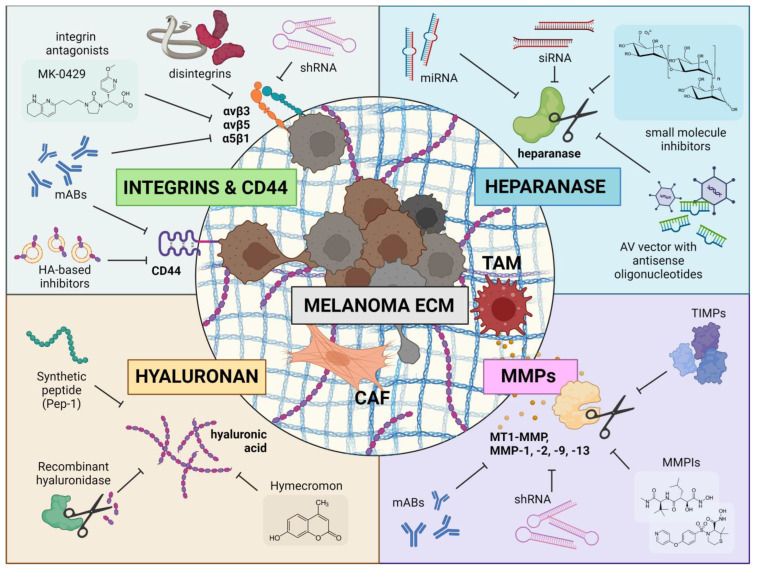
Various approaches to inhibit functional components of the ECM. Abbreviations: MMPs—matrix metalloproteinases, MT1-MMP—membrane type 1-matrix metalloproteinase, MMP-1, -2, -9, -13—metalloproteinase-1, -2, -9, -13, shRNA—short hairpin RNA, MMPIs—matrix metalloproteinase inhibitors, TIMPs—tissue inhibitors of metalloproteinases, AV—adenovirus, siRNA—small interfering RNA, miRNA—microRNA, mABs—monoclonal antibodies, HA—hyaluronic acid, CAF—cancer-associated fibroblasts, TAM—tumor-associated macrophages, ECM—extracellular matrix. Created using BioRender. Mayasin, Y. (2024) https://BioRender.com/h80w259 (accessed on 12 November 2024).

## Data Availability

Not applicable.

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
