# Peer review of "Extracellular Matrix as a Target in Melanoma Therapy: From Hypothesis to Clinical Trials"

_cells, 2024, doi:10.3390/cells13221917_

Round 1
Reviewer 1 Report
Comments and Suggestions for Authors
In the submitted manuscript, Mayasin Y.P and colleagues review the importance of targeting the different components of the extracellular matrix (ECM) reporting in a fairly comprehensive compendium a wide range of different approaches which target the tumor ECM to block the progression of melanoma.
Overall, the review is well organized and the presence of tables summarizing the published preclinical approaches and the clinical trials is helpful for a better understanding, and provides a more schematic view of the approaches. Only minor issues could be rasised:
Line 115: Referring to citation number 49 the correlation of increased heparanase and worse prognosis is true for a sub group of 46/69 patients. I would state this.
Line 117-118: I would specify that the study is preclinical and there is a 29-fold upregulation not a 30-fold as reported in publication n. 50.
Line 152-153: Referring to publication n.59 there is a discrepancy between the number of patients with stable disease respect to what reported in the manuscript (2 out of 5 instead of 9 out of 15) please amend.
Please always write in vivo and in vitro in Italics.
Table page 15: change signalling with signaling.
Line 64: for a better understanding of the sentence, I suggest to change “…to targeting cancer therapy” with “…to improve cancer therapy”
Line 69: “…while the remainder form de novo” change form with forms
Line 91: remove through.
Lines 111, 228, 460: please change “extracellular matrix” with ECM since you have already used the abbreviation.
Line 209: “The PG500 series includes” change includes with include
Line 261: correct “On the one hand…” remove the
Lines 270, 372, 446, 646: Use the abbreviation MMPs
Lines 268, 310, 363: Use the abbreviation MMPIs
Line 523: instead of “…to inhibiting” use “…to inhibit”
Author Response
Dear reviewer, thank you for your detailed and very pertinent comments on the content of our manuscript. Below is a point-by-point account of the changes made to the publication:
- We agree with you and have provided the correct ratio (lines 206-208).
- We have clarified that the cited study is preclinical and that we are talking about a 29-fold increase in heparanase expression (lines 208-211).
- We have clarified in our manuscript that we are referring specifically to the response in melanoma patients (of which there were 5 in the study, hence the 2 out of 5 ratio). We also reported the original response rate for all patients (9 out of 15) based on your comment (lines 244-245).
- We have corrected the spelling of all in vivo and in vitro to italics.
- We have corrected signalling to signaling throughout the text.
6-14. We have made your recommended substitution throughout the text of the article.

Reviewer 2 Report
Comments and Suggestions for Authors
This review manily combined the detailed results of different approaches targeting extracellular matrix elements in melanoma in preclinical and clinical studies.Some points should be noted as below,
1) A recent paper (https://pubmed.ncbi.nlm.nih.gov/37056571/) proposes that cancer can be regarded as a pathological ecosystem in which the neoplastic cells together with tumor microenvironment such as extracellular matrix, and immune cells, and the “mulberry-fish-ponds model” to elucidate the dynamic reciprocity of mutualism between cancer cells and TME (e.g. CAFs synthesizing ECM components), it should be helpful for understanding the the relationship of TME and cancer cells in “1. Introduction”.
2) How about ECM physical traits including stiffness effects on melanoma and its possible targeted therapy.For example, any reports show that melanoma micrometastases display ECM softening and macrometastases are stiffer?
3) The references cited in Table 1-2, in these experiments, are there any that involve 3D cell culture?
Author Response
Dear reviewer, thank you for your detailed and very pertinent comments on the content of our manuscript. Below is a point-by-point account of the changes made to the publication:
- We have gone through the material on the topic of the cancer pathological ecosystem model and included the publication you provided in the review for broader disclosure of the relationship between TME and malignant cells (lines 61-67).
- In the original manuscript text, we briefly described the functional and physical changes of the altered ECM, including increased stiffness and consequent drug resistance (e.g. via the ROR1 and ROR2 signaling pathways). At your suggestion, we have expanded this section to include information on the effects of stiffness on melanoma cells themselves and directly on targeted therapy. We have also added reports on local changes in the ECM during macro- and micrometastasis (lines 162-187).
- To our regret, experiments with canonical 3D models, including spheroids and organoids, are not included in the references cited in Tables 1-2. This is due to the fact that most work with 3D melanoma cultures emphasizes the testing of chemotherapy or targeted drugs directly against the tumor core rather than TME cells when present in co-culture or ECM remodeling (Vörsmann, 2013; DOI:10.1038/cddis.2013.249). Nevertheless, we would like to mention in our publication references to papers [98], [119], [164] that use cell lines cultured in Matrigel matrix, but these papers focus more on tracking migration and invasiveness of tumor cells rather than assessing the dynamics of ECM changes under 3D structure.

Reviewer 3 Report
Comments and Suggestions for Authors
The authors peformed a review about extracellular matrix as a target in melanoma therapy. The article is of interest and well written, however some changes are needed:
- please add a section regarding the role of extracellular vescicles in malignant melanoma and in extracellular matrix in melanoma. Indeed extracellular vescicles (e.g. exosomes and microvescicles) are playing always a more important role in melanoma progression.
- since it is knownw that there are several different type of melanomas (e.g. superficial spreading melanoma, desmoplastic melanoma, spitz melanoma etc etc...) add a paragraph in which you explain how the extracellular matrix can changes according to these different type of melanomas and the relative possible future therapeutic options to target these different types of melanoma.
- stress the importante of neo-adjuvant and adjuvant therapeutic options for melanoma that with the future will be always more evaluated and discuss on how these therapeutic options may target the extracellular matrix.
- add some periods about the role of the extracellular matrix in the pre-metastatic niche formation.
Thank you.
Author Response
Dear reviewer, thank you for your detailed and very pertinent comments on the content of our manuscript. Below is a point-by-point account of the changes made to the publication:
- We have added a block on the role of extracellular vesicles in malignant melanoma and the relationship between EVs and extracellular matrix in melanoma (lines 75-90).
- The authors agree with you on the relevance of information on qualitatively different ECM changes depending on the type of melanoma, but to date no such detailed comparative histological papers have been published. Based on your comments, we have added a separate paragraph about a detailed immunohistochemical comparison of ECM characteristics in different types of pigmented skin lesions, including melanoma in situ, invasive and malignant melanoma (lines 124-148).
- We also fully agree with the importance of mentioning the use of neoadjuvant and adjuvant therapy against melanoma. To our regret, none of the drugs used in such treatments currently target the ECM, but the importance of this target for adjuvant and neoadjuvant therapies is emphasized in the literature, which we have also pointed out in the corrected version of our review (lines 43-56).
- We have added a block on the role of the extracellular matrix in pre-metastatic niche formation, including examples of how this occurs in melanoma (lines 101-113).

Round 2
Reviewer 3 Report
Comments and Suggestions for Authors
The authors improved the manuscript and it can accepted now for publication. Thank you